The EMBO Journal (2013) 32, 1334–1343
www.embojournal.org

THE
EMBO
JOURNAL

# Architecture of the Pol III–clamp–exonuclease complex reveals key roles of the exonuclease subunit in processive DNA synthesis and repair

## Ana Toste Rêgo[1], Andrew N Holding[1], Helen Kent and Meindert H Lamers*

MRC Laboratory of Molecular Biology, Cambridge, UK

**DNA polymerase III (Pol III) is the catalytic α subunit of the bacterial DNA Polymerase III holoenzyme. To reach maximum activity, Pol III binds to the DNA sliding clamp β and the exonuclease ε that provide processivity and proofreading, respectively. Here, we characterize the architecture of the Pol III–clamp–exonuclease complex by chemical crosslinking combined with mass spectrometry and biochemical methods, providing the first structural view of the trimeric complex. Our analysis reveals that the exonuclease is sandwiched between the polymerase and clamp and enhances the binding between the two proteins by providing a second, indirect, interaction between the polymerase and clamp. In addition, we show that the exonuclease binds the clamp via the canonical binding pocket and thus prevents binding of the translesion DNA polymerase IV to the clamp, providing a novel insight into the mechanism by which the replication machinery can switch between replication, proofreading, and translesion synthesis.**

*The EMBO Journal* (2013) **32,** 1334–1343. doi:10.1038/emboj.2013.68; Published online 2 April 2013
Subject Categories: genome stability & dynamics; structural biology
Keywords: crosslinking; DNA polymerase III; DNA sliding clamp; exonuclease; translesion synthesis

## Introduction

The *E. coli* DNA polymerase III holoenzyme (DNA Pol III HE) is a large macromolecular machine that contains 10 proteins (α, β, ε, θ, τ, γ, δ, δ', χ, and ψ) with a combined molecular weight close to one megadalton (Johnson and O'Donnell, 2005). Its catalytic centre is the α subunit, the third DNA polymerase identified in *E. coli*, termed as Pol III (Gefter *et al*, 1971). In isolation Pol III is a rather inefficient enzyme compared to other DNA polymerases. It has a low affinity for DNA and as a result the isolated Pol III is only capable of synthesizing short stretches of DNA (Fay *et al*, 1981; Bloom *et al*, 1997). The processivity of Pol III is greatly enhanced upon binding to the DNA sliding clamp (β subunit) a ring-shaped molecule that encircles the DNA (LaDuca *et al*, 1986;

*Corresponding author. MRC Laboratory of Molecular Biology, Francis Crick Avenue, Cambridge Biomedical Campus, Cambridge CB2 0QH, UK. Tel.: +44 (0)1223 267063; Fax: +44 (0)1223 268305;
E-mail: mlamers@mrc-lmb.cam.ac.uk
[1]These authors contributed equally to this work.

Stukenberg *et al*, 1991). In addition, the exonuclease (ε subunit) is also required for optimal activity (Studwell and O'Donnell, 1990; Kim and McHenry, 1996a). Once assembled into the holoenzyme, Pol III transforms into a highly efficient enzyme synthesizing DNA with a remarkable speed of up to 1000 bp/s and > 80 000 bp synthesized per binding event (Georgescu *et al*, 2012). As a result, during replication a single holoenzyme is sufficient to complete the entire genome (McInerney and O'Donnell, 2004; Reyes-Lamothe *et al*, 2010). However, due to the opposite polarity of the two DNA strands, synthesis at the lagging strand is discontinuous, requiring repositioning of Pol III every 1–3 kb (McInerney *et al*, 2007; Georgescu *et al*, 2012). In addition, errors introduced during DNA synthesis are removed by the exonuclease subunit (Scheuermann *et al*, 1983) upon which Pol III temporarily releases the DNA. Furthermore, chemically modified bases form a block for the high fidelity Pol III and require the action of translesion DNA polymerases. These low fidelity DNA polymerases are capable of DNA synthesis over the lesion after which normal DNA replication can resume (Sutton and Walker, 2001). Interestingly, some controversy has arisen over the switching of the replicative polymerase Pol III and translesion polymerase Pol IV at the site of a lesion. Initial reports suggested a model in which the DNA sliding clamp functions as a 'molecular toolbelt' than can bind two polymerases simultaneously (Indiani *et al*, 2005; Furukohri *et al*, 2008; Wagner *et al*, 2009). However, more recent work challenges the toolbelt model and instead suggests that only one polymerase can bind to the clamp at one point and that therefore Pol IV directly competes of Pol III (Heltzel *et al*, 2009). Regardless of this, it is clear that while on one hand Pol III needs to bind efficiently to the clamp and exonuclease in order to synthesize long stretches of DNA, it also needs to be able to dissociate from the DNA frequently during the different stages of the replication and repair. The mechanisms by which these switches operate are currently not known.

The interactions between the Pol III, clamp, and exonuclease have been studied extensively and atomic structures have been solved for each of the individual subunits (Kong *et al*, 1992; Hamdan *et al*, 2002; Lamers *et al*, 2006). However, no structural information is available on how Pol III interacts with the other two subunits. Therefore, to structurally analyse the interactions between Pol III, clamp, and exonuclease, we made use of chemical crosslinking combined with mass spectrometry to map the interaction sites between the three proteins. We find that in addition to the known direct interaction between Pol III and clamp, the exonuclease provides a second, indirect interaction to the clamp. By doing so, it enhances the interaction between Pol III and clamp and provides the exonuclease with a more efficient access to the DNA. In addition, we find that by occupying the second binding pocket of the clamp the

exonuclease prevents binding of the translesion DNA polymerase Pol IV, providing the first structural insight into the control of translesion synthesis in bacteria.

## Results

### *The exonuclease enhances binding between Pol III and clamp*

DNA Pol III is not a processive enzyme and depends on the sliding clamp to enhance its processivity during DNA synthesis (Stukenberg *et al*, 1991). However, the affinity between Pol III and clamp is rather low, with reported values for $K_D$ of $\sim 1\,\mu$M (Kim and McHenry, 1996b; Dohrmann and McHenry, 2005; Lamers *et al*, 2006), which contrasts with the observation that once assembled into the holoenzyme Pol III can synthesize DNA of $>80$ kb per binding event (Georgescu *et al*, 2012). It has been reported that in addition to the clamp the exonuclease also has an effect on the processivity of Pol III (Studwell and O'Donnell, 1990; Kim and McHenry, 1996a). Therefore, we wondered if the exonuclease could affect the interaction between Pol III and the clamp. To further investigate this, we used analytical size-exclusion chromatography to analyse the interactions between the three proteins (Figure 1A). (Chromatograms of the individual proteins as well as molecular weight standards are shown in Supplementary Figure S1A). Different concentrations (1.5–10 µM) of dimeric Pol III–clamp complexes and trimeric Pol III–clamp–exonuclease were injected onto a Superdex 200 column and small fractions collected during elution of the protein. The fractions were analysed by SDS gel electrophoresis and protein band intensities measured (Figure 1B–D). At low concentration (1.5 µM), Pol III and clamp migrate independently on the size-exclusion column due to the weak interaction between the two proteins. At higher concentrations (5 µM), almost half of the clamp co-migrates with the Pol III, while at 10 µM the two proteins form

an almost single complex. Strikingly, in the presence of the exonuclease (Figure 1, bottom row) the interaction between Pol III and clamp is enhanced, as shown by the co-migration of the clamp in the Pol III–exonuclease complex even at the lowest concentration (1.5 µM). At higher concentrations (5–10 µM), the complex is further enhanced, ultimately creating a robust complex. We also tested the effect of θ, a small non-essential protein that binds to the exonuclease. We find that the addition of θ has no effect on the migration pattern of the Pol III–clamp–exonuclease complex even though it co-migrates with the complex (Supplementary Figure S1B).

To obtain an estimate of the affinity between the different complexes, we fitted a 'sum of two Gaussians' to the migration profile of the clamp (see Supplementary Figure S1C). From the ratio of the volume of the two Gaussians, representing bound and unbound clamp, we can calculate a $K_D$ value using the equation $K_D = [\text{Pol}][\text{Clamp}]/[\text{Pol-Clamp}]$. Additionally, we applied a correction factor for the $\sim$five-fold dilution that takes place on the column (loading volume 50 µl, elution volume 250 µl). Doing so, we find that the $K_D$ for Pol III–Clamp is $1.2 \pm 0.2\,\mu$M, which is similar to values observed before (Kim and McHenry, 1996b; Dohrmann and McHenry, 2005; Lamers *et al*, 2006). Addition of the exonuclease reduces the $K_D$ $\sim$four-fold to $0.3 \pm 0.1\,\mu$M. Hence, this shows that the exonuclease indeed stabilizes the Pol III–clamp complex and provides an explanation for its stimulating effect on processive DNA synthesis (Studwell and O'Donnell, 1990; Kim and McHenry, 1996a).

### *Chemical crosslinking maps the interactions between Pol III, clamp, and exonuclease*

To define the organization of the trimeric Pol III–clamp–exonuclease complex in more detail, we employed a chemical crosslinking approach similar to as described before (Leitner *et al*, 2010). Individual proteins or different protein complexes were incubated with the lysine

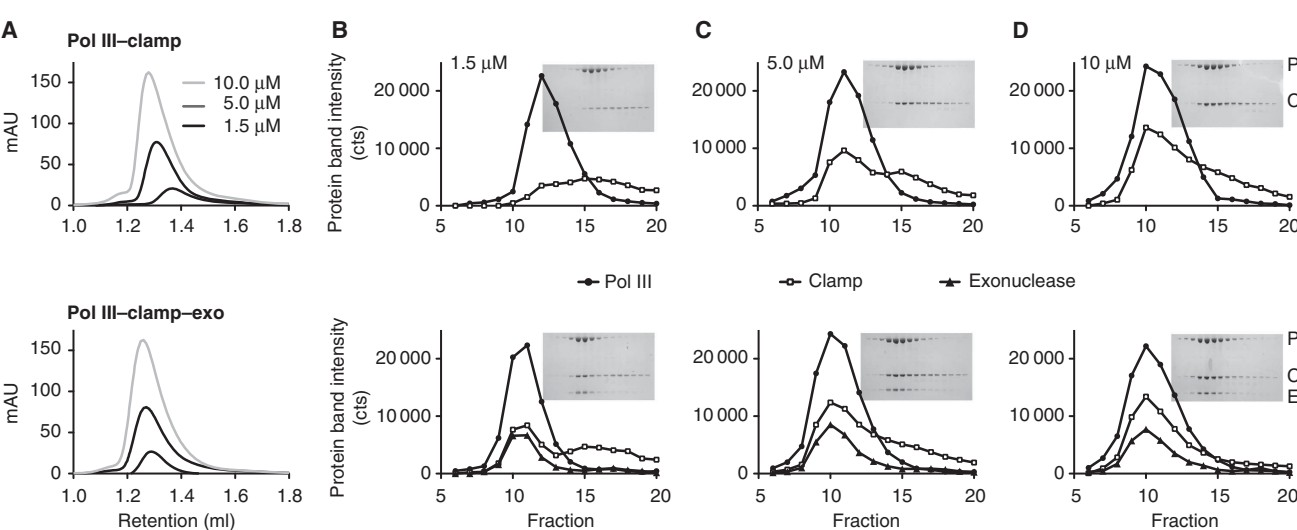

**Figure 1** Exonuclease enhances binding between DNA Pol III and clamp. (**A**) Gel filtration analysis of dimeric Pol III–clamp complex (top panel) and trimeric Pol III–clamp–exonuclease complex (bottom panel) at different concentrations. The retention volume of the Pol III–clamp complex shifts towards higher molecular weight with increasing concentration, due to the association of the two proteins. In contrast, the retention volume of the Pol III–clamp–exonuclease does not change, showing that the complex is stable at all three concentrations. (**B–D**) SDS–PAGE analysis of sequential fractions from the different gel filtration runs shown in (**A**). Filled circles: *P*ol III, open squares: *C*lamp, and triangles: *E*xonuclease. The insert shows the protein gel. At low concentrations (1.5 µM), Pol III and clamp run almost independently of each other (left panel), while addition of the exonuclease (bottom row) causes the clamp to co-migrate with the polymerase. At increasing concentrations, most of the clamp migrates with Pol III (top row). Addition of the exonuclease further enhances the binding (bottom row).

**Table I** Intra and inter protein crosslinks in different complexes

| Crosslink Residue 1 | Residue 2 | Protein complex[a] | | | | |
|---|---|---|---|---|---|---|
| | | P | PE | PC | PCE | Total |
| Pol | Pol | | | | | |
| 29 | 714/5/6 | 16 | 10 | 4 | 8 | 38 |
| 29 | 722 | 10 | 3 | 0 | 5 | 18 |
| 229 | 1009 | 3 | 0 | 1 | 0 | 4 |
| 316 | 595 | 1 | 0 | 0 | 0 | 1 |
| 439 | 1009 | 1 | 1 | 0 | 0 | 2 |
| 461 | 1009 | 3 | 1 | 0 | 1 | 5 |
| 500 | 510 | 1 | 0 | 0 | 0 | 1 |
| 500 | 1009 | 9 | 1 | 0 | 0 | 10 |
| 510 | 1009 | 1 | 0 | 0 | 0 | 2 |
| 617 | 983 | 2 | 0 | 0 | 0 | 2 |
| 621 | 983 | 7 | 1 | 0 | 0 | 8 |
| 621 | 992 | 1 | 0 | 0 | 0 | 1 |
| 855 | 872 | 14 | 6 | 1 | 3 | 24 |
| 983 | 992 | 2 | 0 | 0 | 2 | 4 |
| Exo | Pol | | | | | |
| 120 | 1009 | — | 1 | — | 0 | 1 |
| 136 | 229 | — | 1 | — | 1 | 2 |
| 136 | 510 | — | 1 | — | 0 | 1 |
| 136 | 1009 | — | 2 | — | 1 | 3 |
| 141 | 229 | — | 1 | — | 1 | 2 |
| 141 | 1009 | — | 1 | — | 4 | 5 |
| 158 | 1009 | — | 3 | — | 0 | 3 |
| 235 | 29 | — | 4 | — | 4 | 8 |
| 235 | 714 | — | 4 | — | 2 | 6 |
| Clamp | Pol | | | | | |
| 277 | 872 | — | — | 2 | 6 | 8 |
| 277 | 1009 | — | — | 1 | 4 | 5 |
| Clamp | Exo | | | | | |
| 277 | 136 | — | — | — | 5 | 5 |
| 277 | 141 | — | — | — | 2 | 2 |

Numbers reflect total number of independent crosslinks found over multiple experiments. A dash (—) indicates not applicable.
[a]Polymerase (P), Exonuclease (E), and Clamp (C).

crosslinker bis(sulphosuccinimidyl) glutarate ($BS^2G$) or bis(sulphosuccinimidyl) suberate (BS3) after which the samples were purified by size-exclusion chromatography to remove any non-specific crosslinked products (Supplementary Figure S2A). Next, the purified crosslinked samples were double digested with trypsin and Glu-C protease, fractionated by cation exchange chromatography and analysed by nano-scale reversed phase liquid chromatography coupled to a tandem mass spectrometer for detection and identification of crosslinked peptides. All mass spectra were analysed with an in-house developed program 'Crosslinker' (Andrew N. Holding, manuscript in preparation). This resulted in a total number of 27 unique crosslinks, with the majority of crosslinks measured multiple times (Table I). A detailed list of crosslinked peptides as well as fragmentation spectra are given in Supplementary Figure S2B and C.

For the isolated polymerase (Figure 2A and B), we find a good correlation between the crosslinks and the known crystal structure of *E. coli* Pol III (residues 1–917) (Lamers *et al*, 2006). Notably, a strong cluster of crosslinks is present between residues 29 and residues 714/715/716. Although these are distant in sequence, they are in close vicinity of each other in the protein structure, showing that the crosslinking accurately represents the structure. The average observed distance between the Cα atom of two crosslinked lysines is ~22 Å, which is well within the predicted distance of 24 Å (2 × length of a lysine side chain (6.4 Å) + length of the crosslinker BS3 (11.4 Å)). We also find a few longer crosslinks, with some distances reaching 28 Å. However, these distances are measured on a static crystal

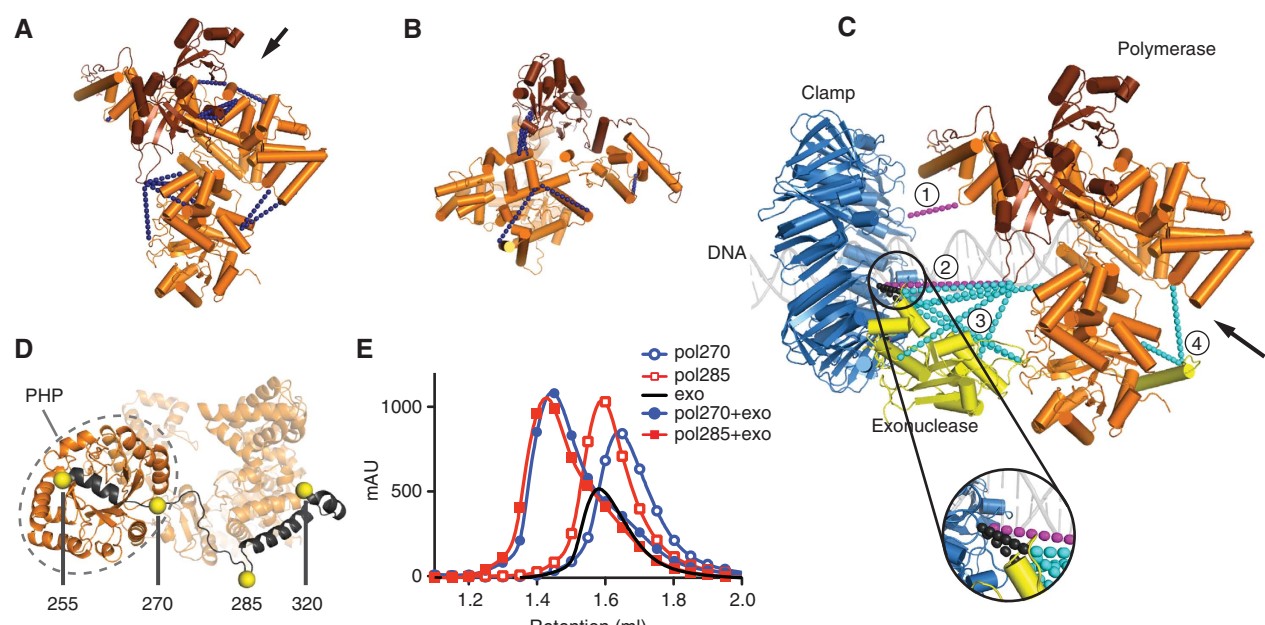

**Figure 2** Chemical crosslinking indicates that the exonuclease is sandwiched between Pol III and clamp. (**A**) Top view of Pol III with crosslinks in blue dashed lines. Arrow indicates view point in (**B**). (**B**) Front view showing the crosslinks between the known part of Pol III and the modelled tail in dark brown. (**C**) Top view of the model of the Pol III–clamp–exonuclease complex, with close-up view of clamp–exonuclease crosslinks (in circle). Dashed lines indicate inter-protein crosslinks. Cyan: Pol III–exo, magenta: Pol III–clamp, black: clamp–exo (internal Pol III crosslinks not shown). See text for more details. Arrow indicates view point for (**D**). (**D**) Bottom view of Pol III (viewed along arrow in **C**). Truncations indicated with yellow spheres, residues 255–320 in dark grey, PHP domain in grey circle, and remainder of polymerase (residues 321–917) in light orange. (**E**) Both N-terminal fragments of Pol III (residues 1–285 and 1–270) shift upon binding of the exonuclease. Pol III[1–270]: blue open circles, Pol III[1–285]: red open squares, exonuclease: solid black line, Pol III[1–270] + exonuclease: solid blue circles, Pol III[1–285] + exonuclease: red solid squares.

structure and are not taking into account any molecular motions of the protein, which are substantial in Pol III and DNA polymerases in general (see Steitz and Yin, 2004; Evans *et al*, 2008; Wing *et al*, 2008). Similarly, distances of up to 28 Å were also observed in Chen *et al* (2010). Interestingly, we also find crosslinks to the tail domain of Pol III (residues 918–1160) that was not included in the crystal structure of *E. coli* Pol III (Lamers *et al*, 2006). To visualize these crosslinks, we created a model of the *E. coli* Pol III tail with the program 'Modeller' (Eswar *et al*, 2006) using the crystal structure of full-length *Thermus aquaticus* Pol III (Bailey *et al*, 2006) as a template. The generated model fits well with the crosslinks that we find (Figure 2A and B), suggesting that the tail of Pol III adopts a similar position in both *E. coli* and Taq Pol III.

In addition to the internal Pol III crosslinks, we also find a large number of crosslinks between Pol III and clamp, Pol III and exonuclease, as well as crosslinks between clamp and exonuclease (see Table I). To visualize these crosslinks, we created a model of the Pol III–clamp–exonuclease complex using the obtained crosslinks as a guide (Figure 2C). Between Pol III and clamp we find two crosslinks (coloured magenta). The first crosslink (labelled with '1' in Figure 2C) brings the internal clamp binding motif of Pol III (residues 920–924) close to the canonical binding pocket of the clamp. This fits well with the previously reported role of the internal motif of Pol III that is essential for the interaction between the two proteins (Dohrmann and McHenry, 2005). The second crosslink between Pol III and the clamp (labelled with '2') positions the clamp in line with the exit path of the DNA (shown in light grey). Between Pol III and the exonuclease, we find two clusters of crosslinks (coloured cyan) separated by >60 Å. The first cluster (labelled with '3') is between the catalytic domain of the exonuclease (residues 1–180) and the 'polymerase and histidinol phosphatase' (PHP) domain (Aravind and Koonin, 1998) of Pol III (residues 1–270). The second cluster of crosslinks (labelled with '4') places the very C-terminus of the exonuclease at the other side of PHP domain, thus wrapping its tail around the polymerase. In addition, and most interestingly, we also find two crosslinks between clamp and exonuclease (coloured black, see zoom). This firmly places the catalytic domain of the exonuclease between the PHP domain of Pol III and the clamp.

Hence, our crosslinking data provide a first structural view of the trimeric Pol III–clamp–exonuclease complex. It confirms the known interactions between Pol III and clamp, but also reveals for the first time how the exonuclease binds to Pol III by wrapping the exonuclease tail around the PHP domain. Furthermore, the crosslinking results suggest a potential direct interaction between the exonuclease and the clamp. These interactions between the exonuclease and the polymerase and clamp are further analysed below.

The location of the exonuclease next to the Pol III PHP domain correlates well with our finding that the first 270 residues of the polymerase are sufficient for exonuclease binding. Previously, it was reported that the first 320 residues of Pol III are sufficient for exonuclease binding (Wieczorek and McHenry, 2006). Yet, this region stretches across the entire length of Pol III and therefore does not provide a detailed map of the exonuclease binding site (Figure 2D). Therefore, we made three additional truncations at residues 255, 270, and 285. The first truncation renders the protein insoluble and

could therefore not be purified. In contrast, the truncations to residues 270 and 285 yield well-behaved proteins that retain full exonuclease binding (Figure 2E). Taken together, our findings show that the tail of the exonuclease binds to the PHP domain of Pol III by wrapping itself around it and placing the catalytic domain adjacent to the exit path of the DNA. This position is similar to the position of the exonuclease domain in the homologous Pol C from *Geobacillus kaustophilus* (Evans *et al*, 2008), but different from the position of the exonuclease domain in Pol I (Beese *et al*, 1993) and Pol II (Wang and Yang, 2009; Supplementary Figure S3).

### Exonuclease binds the clamp using a canonical clamp binding motif

The location of the exonuclease catalytic domain also places it in an ideal position to interact with the clamp. Indeed, upon closer examination, we find a short sequence immediately downstream of the catalytic domain (QTSMAF, residues 182–187) that form a canonical clamp binding motif (Qxx(L/M)xF) found in other proteins that bind the clamp such as Pol II, Pol III, Pol IV, and others (Dalrymple *et al*, 2001). The position of this motif immediately after the catalytic domain fits well with the model predicted from the crosslinking results. The clamp binding motif can easily be modelled into the binding pocket of clamp without violating any of the crosslinking results (i.e., all Cα-Cα distances of crosslinked lysines are kept within 28 Å). Next, to verify that this is indeed a *bona fide* binding motif, we mutated the two conserved hydrophobic residues methionine 185 and phenylalanine 187 to alanine. Following this, we first tested the direct interaction between the exonuclease and the clamp (Figure 3). This interaction is rather weak as even at 30 μM only a fraction of the exonuclease co-migrates with the clamp. In contrast, the mutant exonuclease[185/187] has lost all affinity for the clamp and travels unaltered at all three concentrations, providing proof that the exonuclease binds to the clamp using a canonical binding motif.

### The exonuclease–clamp interaction is required for optimal proofreading activity

To further investigate the role of the direct interaction between exonuclease and the clamp, we made use of a real-time primer extension assay (Song *et al*, 2009). In this assay, the fluorescence intensity of a carboxyfluorescein (FAM) dye at the 5′ end of the template strand is strongly reduced through the extension of the primer strand (Figure 4A). In the presence of Pol III, the signal is rapidly reduced, which is modestly enhanced by the presence of the clamp or the exonuclease alone (Figure 4B). Interestingly, when the exonuclease is present, the fluorescence quickly returns back to starting values due to the activity of the exonuclease that can remove the primer strand once the polymerase runs out of nucleotides. Importantly, the return rate is faster when both exonuclease and clamp are present, while mutation of the clamp binding motif in the exonuclease (M185A + F187A) abolishes the stimulation to levels identical to exonuclease alone. Hence, the direct interaction between the clamp and the exonuclease does not only enhance the interaction between the Pol III and the clamp, but also positions the exonuclease in a conformation that is more favourable for DNA access. This may be explained by the observation that the catalytic domain of the exonuclease is tethered via a

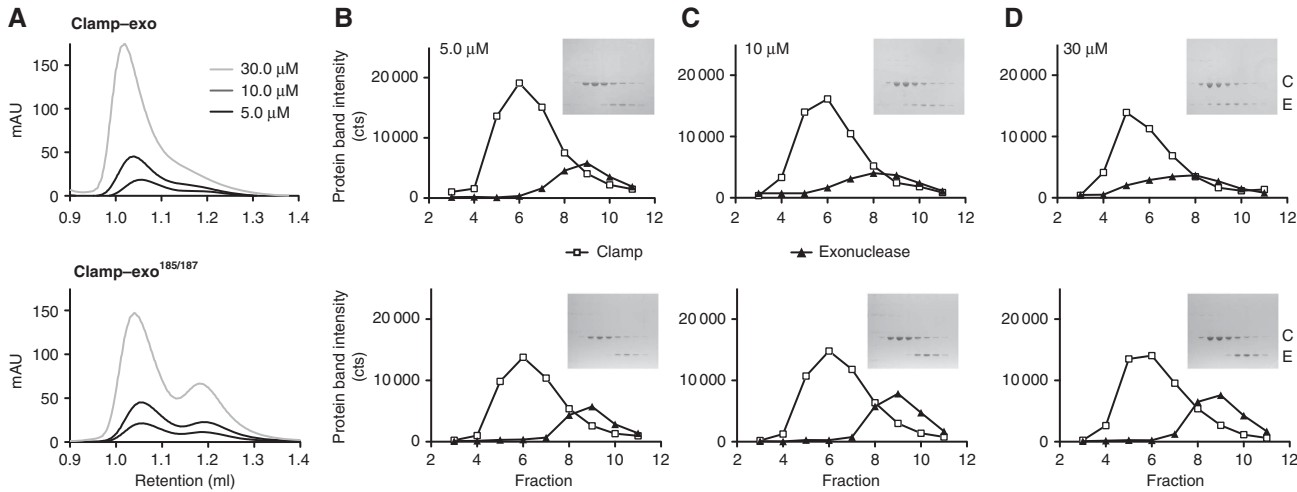

**Figure 3** Exonuclease uses the canonical clamp binding motif to bind the clamp. (**A**) Gel filtration analysis of clamp binding by wild-type exonuclease (top panel) and a mutant exonuclease (bottom panel) where two conserved residues (Met 185 and Phe 187) in the canonical clamp binding motif have been changed to alanine. The retention volume of the wild-type exonuclease–clamp complex shifts towards higher molecular weight with increasing concentration due to the association of the two proteins. In contrast, even at the highest concentration of mutant exonuclease[185/187] clamp and exonuclease migrate in two separate peaks. (**B–D**) SDS–PAGE analysis of sequential fractions from the different gel filtration runs shown in (**A**). Open squares: *C*lamp and triangles: *E*xonuclease. The insert shows the protein gel. With increasing concentrations, the exonuclease co-migrates with the clamp. The mutant exonuclease[185/187] shows no interaction with the clamp and migrates separate from the clamp even at 30 μM protein concentration.

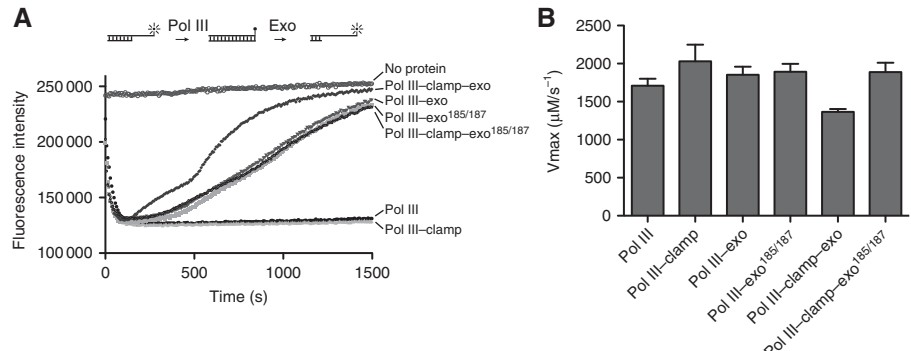

**Figure 4** The clamp binding motif of the exonuclease is important for optimal proofreading activity. (**A**) Real-time primer extension assay in which the intensity of a 5′ placed fluorophore is quenched by the incorporation of nucleotides in the bottom strand (schematically drawn on top). Upon addition of the exonuclease, the fluorescence intensity rapidly returns to starting values due to the removal of nucleotides by the exonuclease. Addition of the clamp to the Pol III–exonuclease complex increases the effectiveness of the exonuclease activity. Mutation of the exonuclease clamp binding motif (exonuclease[185/187]) abolishes this stimulation. (**B**) Quantification of the polymerase activity in the presence of clamp and exonuclease. Vmax was calculated from a titration of nucleotide from 0 to 27 μM, using the initial linear section of the curve only (typically first 30 s) (see Supplementary Figure S5). Displayed values and standard deviations were calculated from three repeat experiments with 12 nucleotide concentrations each.

flexible linker to the polymerase (Ozawa *et al*, 2008) and may therefore not be always in a suitable position to bind the 3′ end of the DNA. Yet, when the clamp is present and limits the movement of the exonuclease it can hold the exonuclease in a position closer to the DNA substrate.

### The tail of Pol III has multiple interactions with the clamp

Previously, it has been reported that Pol III has two clamp binding motifs ($Qx_2[L/M]x_{0/1}F$) that are located at either end of the tail of Pol III: an internal clamp binding motif spanning residues 920–924 (Dalrymple *et al*, 2001; Dohrmann and McHenry, 2005) and a second motif located at the very C-terminus (residues 1154–1159) (López de Saro *et al*, 2003; Georgescu *et al*, 2008). However, as the clamp is composed of two β monomers, there are only two binding pockets per clamp, not enough to bind the two potential binding motifs

from Pol III and a third motif from the exonuclease. It has been shown that the C-terminal clamp binding motif is not required for replication (Kim and McHenry, 1996b; Dohrmann and McHenry, 2005). Here too, we find that the C-terminal binding motif of Pol III does not appear to contribute to clamp binding: we find crosslinks between the clamp and the internal clamp binding motif of Pol III, but none to the C-terminal clamp binding motif. Further evidence that the C-terminal motif does not contribute to clamp binding can be found in the structure of Taq Pol III (Bailey *et al*, 2006), where the motif is buried in between the oligonucleotide/oligosaccharide binding (OB) domain and the C-terminal domain of the polymerase tail and therefore inaccessible for clamp binding (Supplementary Figure S6A and B). Moreover, while the internal clamp binding motif is found conserved in 25 out of 30 bacterial Pol III sequences, the C-terminal motif is only found in 3 species (Supplementary Figure S6C).

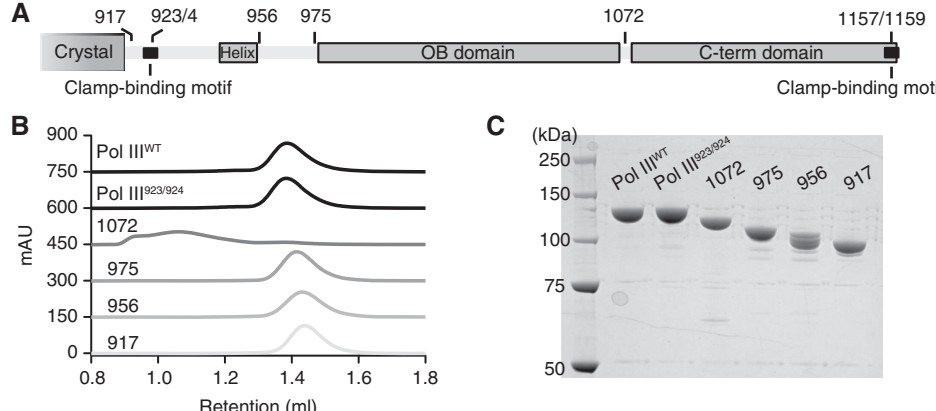

**Figure 5** C-terminal deletions and mutants used in this study. (**A**) Schematic representation of the position of the different C-terminal deletions and mutations. (**B**) Gel filtration analysis of the different proteins. Note the aggregated protein for the deletion at residue 1072. (**C**) SDS–PAGE analysis of the purified proteins.

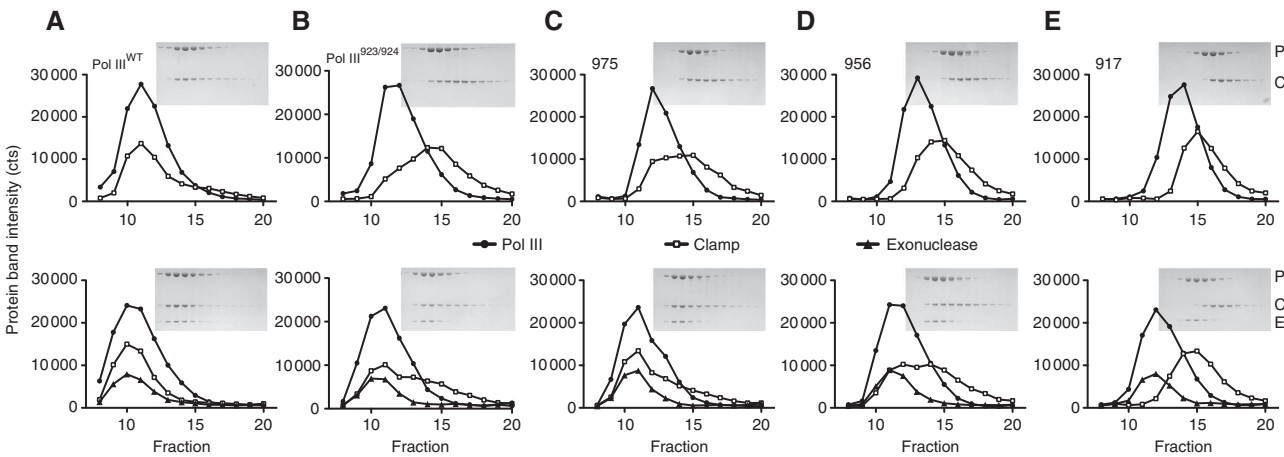

**Figure 6** The tail of Pol III has multiple interactions with the clamp. (**A–E**) Gel filtration analysis followed by SDS–PAGE of the different C-terminal deletion constructs of Pol III in complex with clamp and exonuclease. Increasing deletion of the tail reduces binding to the clamp, indicating multiple interactions between the Pol III tail and the clamp. Addition of exonuclease (bottom row) can partially restore binding for the deletions at residues 975 (**C**) and 956 (**D**), but can no longer do so for the deletion at residue 917 (**E**), as it has lost all clamp binding interaction sites.

Hence, we can expect that in the trimeric Pol III–clamp–exonuclease complex, both binding pockets of the clamp are occupied, one by the internal clamp binding motif of Pol III and one by the exonuclease clamp binding motif. To address the relative contributions the internal clamp binding motif of Pol III and the clamp binding motif of the exonuclease, we mutated the conserved [L/M] and F to an alanine in each motif (i.e., exonuclease M185A + F187A, Pol III M923A + F924A). We already showed that the mutation of the clamp binding motif in the exonuclease results in a loss of clamp binding (Figure 3). Interestingly, mutation of the internal clamp binding motif of Pol III decreased binding (Figure 6B), but does not abolish it, suggesting that the tail of Pol III may have more than one site of interaction with the clamp. Similarly, a mutant of the clamp that has both canonical binding pockets removed still retains Pol III binding, albeit at ∼10-fold lower affinity than the wild-type clamp (Scouten Ponticelli *et al*, 2009).

Therefore to verify if the tail indeed has multiple interactions with the clamp we made a series of deletion constructs delimited by domain structures and potential binding motifs (Figure 5A). All of the deletion constructs could be expressed and purified to homogeneity, except for the deletion at 1072 that renders the protein unstable (Figure 5B) and was therefore excluded from the binding studies. The increasing deletions of the Pol III tail (975, 956, 917) result in a decreasing affinity for the clamp (Figure 6C and D), indicating that the tail has indeed multiple interactions with the clamp. For the deletion constructs 975 and 956 the loss in affinity can be rescued to different degrees by the addition of the exonuclease (Figure 6, bottom row), stressing its importance in clamp binding. The deletion at residue 917 cannot be rescued by addition of the exonuclease, suggesting that this construct has lost all clamp binding regions, as was previously observed in Lamers *et al* (2006). In conclusion, the gradual loss of affinity with the increasing deletions of the tail clearly indicates that the tail of Pol III has multiple interactions with the clamp, with the exonuclease providing an additional contact. Multiple clamp interactions have also been described for other clamp binding proteins: *E. coli* Pol IV (Bunting *et al*, 2003), the archaeal RB69 DNA polymerase (Mayanagi *et al*, 2011), and the T4 phage clamp loader complex (Kelch *et al*, 2011). The nature of the additional Pol III–clamp interactions awaits structural characterization.

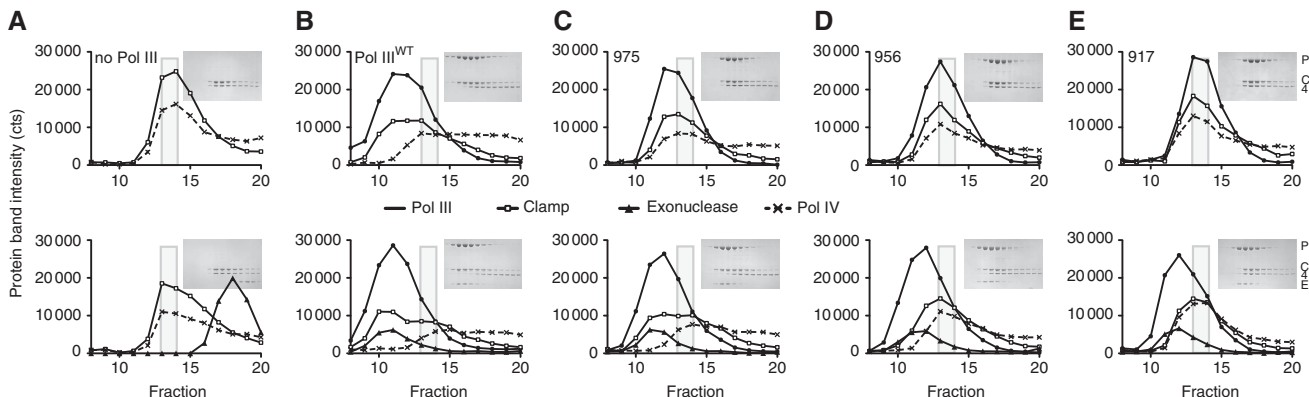

**Figure 7** Exonuclease prevents binding of DNA Pol IV to the clamp. (**A**) Pol IV (dashed line with crosses) readily forms a complex with the clamp (open circles). Addition of exonuclease does not disrupt the complex (bottom panel). Grey box indicates peak fractions 13 and 14 of Pol IV–clamp complex. Protein concentration used: Pol IV 20 μM, clamp 10 μM (dimer), and exonuclease 10 μM. (**B**) Addition of Pol III (10 μM) displaced most of Pol IV from the clamp, which is further enhanced in the presence of the exonuclease (bottom panel). (**C**, **D**) Partial deletion of the Pol III tail enables Pol IV to regain access to the clamp. Note that the migration of Pol IV–clamp complex overlaps with the migration of the Pol III deletions constructs and does not indicate a trimeric complex between Pol III–Pol IV–clamp. Such a trimeric complex would elute at fractions 10 and 11 similar to the trimeric Pol III–clamp–exonuclease complex (**B**). Addition of the exonuclease can restore much of the binding between Pol III and clamp in deletion 975, but only marginally for the deletion at residue 956 (bottom row). (**E**) All binding between Pol III and clamp is lost in the deletion at residue 917, allowing Pol IV full access to the clamp, with the exonuclease unable to rescue binding (bottom panel).

### The Pol III tail and exonuclease prevent binding of the translesion DNA polymerase IV to the clamp

With both binding pockets of the clamp occupied by Pol III and the exonuclease, as well as additional interactions of the Pol III tail with the clamp, most of the clamp is protected in the trimeric Pol III–clamp–exonuclease complex. Yet, it has been reported that Pol III and the translesion (TLS) DNA polymerase Pol IV can bind to the clamp simultaneously (Indiani *et al*, 2005) and that the two polymerases switch positions on the DNA during stalling of the replication machinery, but not during active replication (Indiani *et al*, 2005; Furukohri *et al*, 2008; Heltzel *et al*, 2009; Wagner *et al*, 2009). In addition, Pol IV binds to the clamp using the canonical binding motif (Bunting *et al*, 2003). Hence, we wondered if the exonuclease and the tail of Pol III play a part in the regulation of the access of Pol IV to the clamp. Therefore, we analysed the interactions of the clamp, Pol III, and Pol IV, in absence and presence of the exonuclease (Figure 7). Alone, Pol IV readily forms a complex with the clamp, which is not disrupted by addition of the exonuclease (Figure 7A, bottom panel). In contrast, addition of Pol III effectively competes off Pol IV, even with a two-fold higher concentration of the latter (Figure 7B). Addition of the exonuclease further displaces Pol IV from the clamp (bottom panel). When using increasing deletions of the Pol III tail, Pol IV can access the clamp again (Figure 7C–E). Addition of the exonuclease to the complex displaces Pol IV again in deletion 975, to a lesser extent in the 956 deletion construct while no rescue is seen for the 917 deletion construct (bottom row). Hence, our findings show that the exonuclease subunit plays an important role in preventing access of the Pol IV polymerase to the replication machinery by occupying the second binding pocket of the clamp. In addition, the whole of the Pol III tail (residues 917–1160) is required to bind sufficiently tight to the clamp and compete off Pol IV, providing further evidence that the tail of Pol III has multiple interactions with the clamp. Reversely, it has been found that Pol IV too has multiple contacts to the clamp: one via the canonical clamp binding pocket and a second contact on the side of the clamp (Bunting *et al*, 2003; Heltzel *et al*, 2009).

## Discussion

During replication of the genome, the replicative DNA polymerase Pol III needs to associate tightly with the clamp in order to synthesize very long stretches of DNA. At the same time, Pol III needs to be able to quickly change its position on, or dissociate from the clamp in response to different events such as: (i) handing over the DNA to the exonuclease after incorporation of the wrong nucleotide, (ii) making place for a translesion DNA polymerase upon encountering of the chemically modified base, or (iii) repositioning of Pol III at the end of an Okazaki fragment. Thus, rather than a simple tether to the DNA, the interaction between Pol III and the clamp is complex, requiring a substantial degree of control. This is further exemplified by the different reports on the polymerase switch between the Pol III holoenzyme and Pol IV. At low concentrations, Pol IV appears to bind to the clamp simultaneously with Pol III (Indiani *et al*, 2005; Furukohri *et al*, 2008). Low concentrations of Pol IV appear also to be required for translesion synthesis past nitrofurazone-induced DNA lesions (Wagner *et al*, 2009). At higher concentrations, Pol IV is capable of displacing a stalled Pol III holoenzyme from the primer junction *in vitro*, ultimately resulting in a complete inhibition of replicative DNA synthesis (requiring a 25- to 100-fold molar excess of Pol IV) (Indiani *et al*, 2005; Furukohri *et al*, 2008; Heltzel *et al*, 2009; Wagner *et al*, 2009). Interestingly, intracellular Pol IV levels during normal growth are estimated to be ∼10-fold higher than that of Pol III holoenzyme, while during the SOS response, Pol IV levels increase to ∼100 times that of Pol III levels (Kim *et al*, 2001), while even higher levels of Pol IV result in lethality *in vivo* (Uchida *et al*, 2008). Hence, it appears that Pol IV has potentially two roles, one where it can act in consort with the replication machinery to do 'on the fly repair' whereas at high concentrations during the SOS

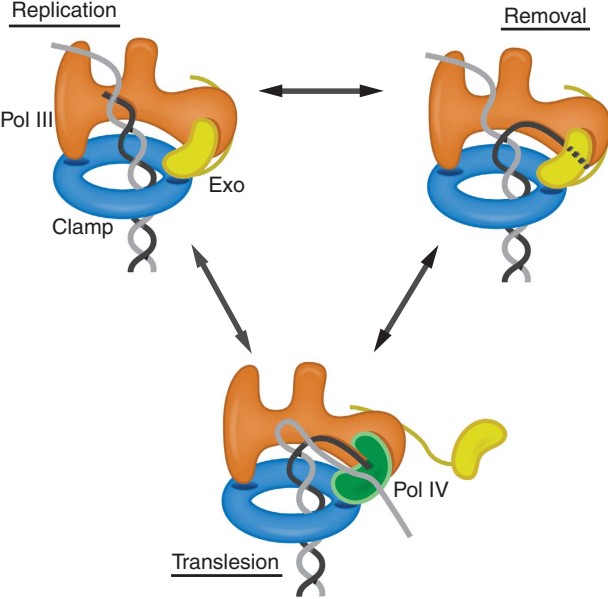

**Figure 8** Model for polymerase switching at the clamp. During normal *replication*, both binding pockets of the clamp are occupied: one by the Pol III directly and one by the exonuclease forming a second, indirect interaction between Pol III and clamp. When Pol III inserts a wrong base, DNA synthesis slows down allowing the exonuclease access to the DNA resulting in *removal* of the wrongly incorporated base. However, when the error in the DNA is on the template strand (in grey), the exonuclease has no access to the lesion (working solely on 3′ end of the DNA). Then, to bypass the lesion, the low fidelity *translesion* DNA polymerase Pol IV that is able to synthesize over the lesion is allowed access to the clamp and DNA. As Pol IV is not very processive, Pol III will ultimately regain access to the DNA and reinitiate high-speed DNA synthesis where the exonuclease repositions itself between Pol III and clamp and thus displacing Pol IV. See also Supplementary Figure S7.

response Pol IV displaces the replication machinery from the DNA template, thus acting as a cellular checkpoint (Uchida *et al*, 2008).

Having a modular system in which multiple sites interact with the clamp allows for a finer control of the binding between Pol III and the clamp. Our observation that the exonuclease forms a second, indirect interaction between Pol III and the clamp provides a simple yet elegant mechanism by which the access of the mutagenic TLS polymerase Pol IV is regulated, which is detailed in Figure 8. During replication, Pol III is tightly tethered to the DNA via the clamp and the exonuclease. When DNA synthesis is blocked upon encounter of a DNA lesion, the replicative polymerase Pol III dissociates from the DNA, due to its intrinsically low DNA binding affinity (see Supplementary Figure S7). When the lesion is in the form of a mis-incorporated base, the exonuclease can simply remove it, after which DNA synthesis can resume. However, when the lesion is in the template strand in the form of a chemically altered base, the exonuclease has no access to the lesion, resulting in a fruitless cycle of DNA synthesis and removal by Pol III and the exonuclease, respectively. This stalling of the replication machinery then enables Pol IV to access the DNA and compete for clamp binding. As the affinity of the exonuclease for the clamp is weaker than that of Pol III (see Figures 1 and 3) it is more likely to dissociate first from the clamp. A flexible linker between the catalytic domain of the exonuclease and its tail (Ozawa

*et al*, 2008) allows the exonuclease to displace itself from the clamp while remaining attached to Pol III. This then enables Pol IV to access the clamp and DNA, allowing it to bypass the lesion in the DNA. Once the lesion has been bypassed, Pol III can regain hold of the DNA and reinitiate high speed DNA synthesis again by displacing Pol IV. Thus, our model predicts that it is the structure of the DNA template rather than the protein itself that dictates which activity is selected: rapid DNA synthesis by Pol III, nucleotide removal by the exonuclease, or lesion bypass by Pol IV.

## Materials and methods

### Materials
All chemicals were purchased from Sigma unless stated otherwise. All reactions were performed in 50 mM Hepes pH 7.5, 150 mM NaCl, and 2 mM DTT.

### Protein purification
All proteins were expressed in *E. coli* BL21 (DE3) or *E. coli* BL21 (DE3) pLyS for Pol IV and purified as described before with some alterations (Maki and Kornberg, 1985; Kong *et al*, 1992; Miller and Perrino, 1996; Beuning *et al*, 2006). In brief, N-terminally His-tagged (N-His$_6$) Pol III (α subunit) was purified using a Histrap, Resource Q, and a Superdex 200 column (all columns from GE Healthcare). N-His$_6$ clamp (β subunit) was purified using a Histrap column, followed by a Superdex 200 column. N-His$_6$ Exonuclease (ε subunit) was purified from inclusion bodies in 6 M Urea using a Histrap column. The protein was refolded by overnight dialysis to 0 M Urea and concentrated using a Resource Q column. Pol IV was purified using a Capto S column, Hitrap Phenyl column, and Superdex 200 column. All proteins were flash frozen in liquid nitrogen and stored at − 80°C. For non-tagged protein, Pol III was purified by Source Q, Heparin, Resource Q, and Superdex 200 columns. Non-tagged clamp was purified using Phenyl sepharose, Hitrap Q, and Hitrap SP columns. Non-tagged exonuclease was obtained after removal of the His$_6$ tag by a 3-h incubation at room temperature with human rhinovirus 3C protease. His-tagged and non-tagged proteins did not show a difference in their migration pattern on a gel filtration column (see Supplementary Figure S4 for comparison).

### Size-exclusion chromatography analysis of exonuclease binding
Samples of the different complexes were prepared at 1.5, 5.0, 10.0, and 30.0 μM and 50 μl injected onto a PC3.2/30 (2.4 ml) Superdex 200 gel filtration column (GE Healthcare) pre-equilibrated in 50 mM Hepes pH 7.5, 150 mM NaCl, and 2 mM DTT. A superdex 75 column was used for the clamp-exonuclease complex (Figure 3). In all, 50 μl fractions were collected and analysed by SDS–PAGE using 4–12% NuPage Bis-Tris precast gels (Life Technologies). Gels were stained with Coomassie brilliant blue and protein band intensities measured using ImageJ (Schneider *et al*, 2012). For estimation of $K_D$ between the clamp and Pol III ± exonuclease, we used only the measured band intensities of the clamp to fit a 'sum of two Gaussians' using GraphPad Prism (version 5 for Mac OSX, Graphpad Software, San Diego, CA, USA). The ratio of the area under the curve of the two Gaussians was then used to calculate the concentration of bound and free clamp, using the starting concentration for total concentration of clamp. A correction factor of 5 was applied to compensate for the dilution of the proteins on the gel filtration column (input volume 50 μl, elution volume 250 μl). Calculated values for $K_D$ are the average of nine experiments (three repeats of three concentrations).

### Real-time DNA primer extension assay
Activity of Pol III was measured using a 38-nt long DNA substrate annealed to a 30-nt primer strand. The carboxyfluorescein moiety was located at the 5′ end of the template strand. Template strand: 5′/56-FAM/-CC CCC CCC CGC ACC TAA AGT TGG GAG TCC TTC GTC CTA-3′. Primer strand: 5′-TAG GAC GAA GGA CTC CCA ACT TTA GGT GC-3′. Reactions were performed by mixing different concentrations of dGTP (from 0 to 27 μM) with 100 nM labelled DNA, and 1 μM unlabelled DNA in a final volume of 20 μl in buffer

10 mM Tris–HCl pH 8.0, 50 mM NaCl, 0.5 mM EDTA, 1 mM DTT, 6 mg/ml BSA. Reactions were initiated by addition of 270 nM protein (Pol III, clamp, exonuclease) and 10 mM $MgCl_2$ (final concentrations) and measured in a 348-well plate using a BMG Labtech Pherastar FS plate reader during 30 min with 7 s intervals at 25°C. Data were analysed in GraphPad Prism and the kinetics was calculated by using the initial linear section of the curve (approximately the first 30 s) as shown in Supplementary Figure S5.

### Crosslinking and mass spectrometry analysis
Protein complexes were crosslinked at 40 μM in 50 μl using either 2 mM bis[sulphosuccinimidyl] glutarate ($BS^2G$) or bis[sulphosuccinimidyl] suberate (BS3) (Pierce) using a 50/50% mix of deuterated (d4) and non-deuterated crosslinking reagent. Reactions were incubated for 15 min at room temperature and quenched with 50 mM $NH_4HCO_3$. Samples were subsequently injected onto a PC3.2/30 (2.4 ml) Superdex 200 gel filtration column (GE Healthcare). In all, 50 μl fractions were collected and analysed by SDS–PAGE using 4–12% NuPage Bis-Tris precast gels (Life Technologies) (Supplementary Figure S2A).

Selected fractions were made up to 100 μl using 100 mM $NH_4HCO_3$ and reduced with 5 μl DTT at 10 mM in 100 mM $NH_4HCO_3$ and alkylated with 4 μl iodoacetamide at 55 mM in 100 mM $NH_4HCO_3$ before digestion with porcine sequencing grade trypsin (Promega) at a protein-to enzyme ratio of 20:1 (w:w) in 3 M Urea/100 mM $NH_4HCO_3$. The digest product was de-salted using Sep-Pak Light tC18 (Waters) as directed by manufacturer and then lyophilized. The resultant peptide mixture resolved into the initial buffer for fractionation by strong ion exchange (SCX) performed using a PolyLC Poly SULPHOETHYL A column (5 μM, 300 Å, 50 mm × 1.0 mm). Peptides were eluted using a linear gradient from 30% acetonitrile in 5 mM $KH_2PO_4$ to 30% acetonitrile in 5 mM $KH_2PO_4$/350 mM KCl over 75 min at 80 μl/min, before sub-digestion with endoproteinase Glu-C (Promega) divided equally between fractions such as the total amount is to a ratio of 20:1 (w:w) to initial protein amount.

Selected fractions were desalted by ZipTip C18 (Millipore) and analysed using a Dionex U3000 HPLC machine coupled to a Thermo-Scientific LTQ-Orbitrap Velos mass spectrometer (Thermo Scientific). The reversed-phase LC separations were performed using a Dionex Acclaim PepMap100 column (C18, 3 μM, 100 Å, 75 μm × 150 mm). Peptides were eluted using a linear gradient from 5% acetonitrile in 0.1% formic acid to 40% acetonitrile in 0.1% formic acid over 110 min at 200 nl/min. A cycle of one full FT scan mass spectrum ($m/z$ 350–1800, resolution of 60 000 at $m/z$ 400, lock mass at 445.120025) was followed by 10 data-dependent MS/MS on the most abundant signals with $Z \geqslant 2+$ acquired in the linear ion trap.

The raw data files were processed into a mascot generic format (mgf) using the Thermo Scientific software Proteome Discoverer v1.0. The mgf files were then imported into Crosslinker, in-house software that searches deconvoluted data for pairs of ions within 120 scans of each other and 4 Da apart (Andrew N Holding, manuscript in preparation). The list of potential mass candidates was then compared with a theoretical *in-silico* digest of the crosslinked Pol III, clamp, and exonuclease sequences and scored by a modified version of the algorithm described by Cox *et al* (2011). The top scoring MS/MS spectra of any MS doublets found to match within 2 p.p.m. of the mass of a theoretical crosslinked peptide were then confirmed manually. See also Supplementary Figure S2B and C.

### Modelling of the Pol III tail
The model of the *E. coli* polymerase tail was generated with the program Modeller (Eswar *et al*, 2006) using the crystal structure of Taq Pol III (Bailey *et al*, 2006) as a template. A sequence alignment of *E. coli* and *T. aquaticus* Pol III used for the modelling was calculated with Clustal (Larkin *et al*, 2007) using Pol III sequences from 35 different bacterial species. The model was further manually adjusted in PyMol (Schrödinger, 2010) and Coot (Emsley *et al*, 2010) using the OB domain from *G. kaustophilus* Pol C (Evans *et al*, 2008). Figure 2A–D was prepared with PyMol.

### Supplementary data
Supplementary data are available at *The EMBO Journal* Online (http://www.embojournal.org).

## Acknowledgements

This work was supported by Medical Research Council Grant U105197143. We would like to thank Claudio Cifferi for helpful suggestions and our colleagues for discussion and critical reading of the manuscript.

*Author contributions*: ATR, ANH, and MHL designed, performed, and analysed experiments, HK performed experiments. ATR, ANH, and MHL wrote the manuscript.

## Conflict of interest

The authors declare that they have no conflict of interest.

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
