## [Review Process File · The EMBO Journal]

Manuscript EMBO-2012-83587

Architecture of the Pol III-clamp-exonuclease complex reveals key roles of the exonuclease subunit in processive DNA synthesis and repair

Ana Toste Rego, Andrew N Holding, Helen Kent and Meindert H Lamers

Corresponding author: Meindert Lamers, Medical Research Council

Review timeline:

Submission date:	11 October 2012
Editorial Decision:	15 November 2012
Revision received:	13 February 2013
Editorial Decision:	26 February 2013
Revision received:	27 February 2013
Accepted:	27 February 2013

Editor: Hartmut Vodermaier

Transaction Report:

1st Editorial Decision

15 November 2012

Thank you for submitting your manuscript for consideration by The EMBO Journal. We have now received the comments of three expert referees in the field, which you will find copied below. While the referees in principle appreciate your approach, as well as your main findings, they however raise a number of substantial concerns that in our view preclude publication of this work, at least in the current form. The main criticisms in this regard pertain to experimental problems, methodological explanations, interpretation and discussion (also in light of the existing literature), as well as the need for more quantitative data. Since these points appear to be addressable in principle, I would nevertheless be inclined to allow you an opportunity to submit a revised manuscript for further considerations; however it is apparent that decisively addressing these points will require substantial further experimental efforts, and it is not fully clear whether this could be achieved within the single round of major revision we normally allow for. I would therefore also understand if you were instead to seek rapid publications without major changes elsewhere. If you do decide on revising the manuscript for The EMBO Journal, please make sure to thoroughly respond to all the points raised at this stage, since we will only be able to consider one single round of re-review, whose outcome will depend on convincing all three referees that their concerns have been adequately clarified. From an editorial point of view, I would also encourage you to rewrite both introduction and discussion sections, to put the work, its rationale and its findings more properly into context.

We generally allow three months as standard revision time, and it is our policy that competing manuscripts published here or elsewhere will have no negative impact on our final assessment of your revised study during this three-months period. However, we request that you contact the editor as soon as possible upon publication of any related work, to discuss how to proceed; and I would also appreciate if you could keep us informed on whether or not you plan to pursue the option of

submitting a revised manuscript and within which time frame.

Thank you for the opportunity to consider this work publication, and please do not hesitate to get back to me should you require further clarification regarding this decision or your revision.

REFEREE REPORTS

Referee #1

In this work the authors show that the epsilon proofreading subunit of DNA polymerase III contacts the beta processivity clamp, which increases the stability of the alpha polymerase/beta clamp complex. The authors postulate that the interaction between epsilon and the beta clamp could prevent PolIV from binding to the clamp during active processive replication. The crosslinking and MS results provide an interesting approach that complements structure determination of the individual subunits in the alpha/beta/epsilon complex, providing a novel view of this central subcomplex within the replicative DNA polymerase III holoenzyme. While the model is generally supported, several concerns listed below need to be addressed to provide the confidence needed to accept the attractive overall model.

Major points:

1. The authors neglect to show any sizing standards for the column chromatography experiments. In addition they only conduct the chromatography experiments on the protein complexes. Independent experiments with both protein sizing standards and individual proteins are necessary to complete these results.
2. The authors posit a direct interaction between the beta clamp and epsilon but do not test this interaction directly with individual purified proteins in the absence of alpha. This would help to rule out an alternative explanation for the results in which epsilon indirectly stabilizes the alpha/beta complex. Such an indirect model persists since only a very small number of crosslinks (2) were found between epsilon and beta.
3. The experiment in Figure 3A shows a very minor difference in the migration of the clamp in the wild-type versus mutant Exo background. The authors should verify that this difference is real using their MS method
4. The authors quantify the protein band intensity from sequential fractions from the column chromatography experiments yet they do not have error bars on their results. Were these experiments done multiple times? Error estimates need to be shown to allow the reader to determine the confidence in the results.
5. The authors do not address whether DNA affects the stability of the interactions. One would expect DAN to enhance the apparent affinity of the polymerase/clamp complex due to localization effects and one wonders whether this would alter the role of epsilon.

Minor points:

1. Figure S1: the legend states that stars indicate fractions used for mass spec analysis but I cannot see stars on the image.
2. Figure S5: It would seem that a replication fork structure would be more appropriate for hte gel shift experiment rather than double stranded DNA.
3. Were the 6xhis tags removed from the proteins prior to the experiments? The authors should make a statement regarding any effects that they might expect from the tags.

Referee #2

Summary: The manuscript by Rêgo et al. probes interactions of the E. coli sliding clamp protein with the alpha (catalytic) and epsilon (proofreading) subunits of the main replicase, DNA

polymerase III (Pol III), and with the specialized translesion synthesis Pol IV. Based on results of gel filtration and crosslinking experiments, the authors conclude that epsilon contacts the cleft of the clamp to regulate access of Pol IV to the replication fork. Although the proposed model is intriguing, the results, as presented, are qualitative rather than quantitative, making it difficult to gauge the significance of the proposed interaction. Related to this, the study also lacks functional studies that could speak to the biological relevance of the epsilon-clamp interaction. In addition, certain of the findings presented are confirmatory of earlier published works, reducing the overall novelty and impact of the work. Finally, the authors fail to discuss their findings in the context of previously published models for Pol III-Pol IV switching.

Major comments:

1. The authors have chosen to use gel filtration chromatography to characterize interactions of epsilon with the alpha and beta subunits of Pol III. Although they present evidence that epsilon interacts with the Pol III alpha-clamp complex, and appears to stabilize it, the results are qualitative in nature. Quantitative measure of the effect that wild type and mutant epsilon proteins have on the interaction are possible, and would help to greatly strengthen the authors' argument that epsilon contacts the clamp directly to stabilize the complex rather than influencing the affinity of Pol III alpha for clamp through an alpha-epsilon interaction. In addition, possible impact of other Pol III subunits (i.e., theta, which interacts with epsilon) on this interaction were not examined. Finally, the functional significance of the epsilon-clamp interaction is not examined. As a result, the importance of this interaction to Pol III function is unclear.

2. The authors characterize the contributions of the "internal" and "C-terminal" clamp binding domains of Pol III alpha, and conclude that only the internal motif is involved in contacting the clamp. As noted by the authors, Dohrmann and McHenry (JMB, 350:228-239) presented compelling evidence that the internal clamp-binding motif of Pol III alpha was required for interaction with the clamp, while the C-terminal motif played an important role in contacting the tau subunit of Pol III. Although relevant to the story, the results presented here in regards to the internal and C-terminal clamp binding motifs are largely confirmatory of previously published works. Furthermore, on page 11, the authors discuss contributions of clamp surfaces other than the cleft to the clamp-Pol III alpha interactions. They may wish to discuss results from Scouten Ponticelli et al. (NAR, 37: 2796-2809), which presented evidence for additional contacts using a mutant clamp lacking functional clefts.

3. On page 12, and in the Discussion, the authors discuss models for Pol III-Pol IV switching, and the impact that the Pol III-epsilon-clamp interactions may have on the switch. However, these discussions do not take into account published models for the switch, including Indiani et al. (Mol. Cell, 19: 805-815), Furukohri et al. (JBC, 283: 11260-11269) and Wagner et al. (Mol. Microbiol., 74: 1143-1151), which proposed classical "tool belt" type models, and Heltzel et al. (PNAS, 106:12664-12669), which proposed a variation of the tool belt model involving a single clamp cleft. Presentation of quantitative and functional analyses of Pol IV interactions with different clamp-Pol III alpha-epsilon complexes would contribute significantly to this discussion.

Minor comments:

1. On page 4, the authors report the interaction between Pol III alpha and the clamp to be on the order of 1-10 uM. However, there are several reports that also conclude the interaction is sub-micromolar, including Kim and McHenry, JBC, 271: 20699-20704; Ponticelli et al., NAR, 37: 2796-2809.

2. On page 7, the authors conclude, based largely on results of cross-linking experiments, that epsilon interacts directly with the clamp. Since these cross-linkers have spacer arms of ~7-12 angstroms, this conclusion strikes this reviewer as an overstatement.

Referee #3

Exonuclease ϵ stabilizes the Pol III - clamp complex and prevents translesion DNA Pol IV access to the clamp

Ana Toste Rêgo, Andrew N. Holding, Helen Kent, and Meindert H. Lamers

In this manuscript the authors describe the structural characterization of a trimeric E.coli Pol III complex composed of the catalytic α , the DNA sliding clamp β and the exonuclease ϵ subunit. The identification of chemically cross-linked lysine residues by mass spectrometry indicated protein domains that are in close vicinity and allowed to delineate a subunit topology. Based on this topology and previous observations the authors generated mutant subunit proteins to narrow down the binding interfaces and tested the effect of these mutations on protein-protein binding by size exclusion chromatography and co-elution of the interacting proteins. Gelfiltration of the wild-type proteins indicated that the exonuclease enhances the binding of the catalytic to the clamp subunit which is in agreement with the cross-link data that position the exonuclease between the catalytic and clamp subunits. Cross-links further suggested that the exonuclease binds the PHP domain of the catalytic subunits by wrapping its C-terminal tail around. Mutations of the clamp binding motifs of the catalytic and exonuclease subunit revealed that these motifs are important for the interaction of the catalytic with the clamp subunit. Truncations of the C-terminal tail of the catalytic subunit revealed that several sites within the C-terminus contribute to clamp binding. In addition, the authors showed that the catalytic subunit of Pol III competes with Pol IV for binding to the clamp subunit and that this effect is enhanced by the presence of the exonuclease. The authors suggested that the various sites of the catalytic and exonuclease subunit contributing to clamp binding provide a mechanism for the fine-tuning of the replication machinery that allows the removal of bases by the exonuclease and the repair of DNA lesions by Pol IV.

In general, the conclusions are in agreement with the experimental findings. It has to be pointed out that cross-links are rather "flexible restraints" that indicate spatial proximity between protein domains and facilitate to draw a topological map at low resolution. Consequently, the findings that cross-links indicate that the exonuclease is sandwiched by the catalytic and clamp subunits or that the exonuclease tail may be wrapped around the catalytic subunit provide a novel structural model that clearly needs further structural analysis by EM or crystallography. The reviewer thus argues for a more careful interpretation of the data by taking into account that a topology based on a few clusters of cross-links may be associated with significant inaccuracies.

Major comments

1. The cross-links identified by mass spectrometry are crucial for the design of the binding experiments and the interpretation of the results. Based on the information shown in the manuscript it is impossible to judge the quality of the cross-linking analysis. The reviewer thus requests detailed information on the cross-link identifications. As a minimum requirement the authors have to provide the sequences of the cross-linked peptides indicating the linked lysines and the precursor mass error. Furthermore, the authors have to state the minimum number of fragment ions per peptide that was considered to be sufficient for confident cross-link identifications. Likewise, the reviewer recommends to provide labeled fragment ion spectra of cross-links that are important for the conclusions of this study.
2. The authors state that the cross-links are in agreement or do not violate the structural model, however, they do not explain what measure they apply to judge violation of the cross-linking results. The authors should explain how they calculate this distance measure and whether there is also some flexibility associated with this value.
3. On page 5 it is indicated that the authors employed a cross-linking approach similar to Chen et al. In contrast to this study, Chen et al. did not use isotopically labeled cross-linkers. With the exception of the strong cation exchange chromatography for the enrichment of cross-linked peptides there are no similarities to the strategy described by Chen et al. In case there are no suitable references describing this approach, the authors have to explain their strategy and the search engine used to identify the cross-links in more detail.

In its present form the manuscript is not recommended for publication in EMBO Journal.

Minor comments:

1. The authors used several times the term "affinity".
On page 4 "the affinity between Pol III and clamp is rather low, between 1-10 μM ": μM rather refers to the KD.
In several other instances the authors used the term "affinity", however, were not measuring affinity but assessed enhanced/decreased binding or interaction based on co-elution from the gelfiltration column.
2. Page 5, line 23: "(Chen et al, 2009)" change to "(Chen et al, 2010)"
3. Page 8, line 24: "extend" change to "extent"

4. Page 11, line 11: The acronym "OB" has not been introduced.
5. Page 12, line 17: "competes of Pol IV" change to "competes off Pol IV"
6. Page 16, line 15: "reverse-phase" change to "reversed-phase"

At the author's/editor's discretion

It remains within the editor's/author's responsibility to decide whether cross-links identified by a software that has not yet been published are acceptable.

1st Revision - authors' response

13 February 2013

Referee #1

In this work the authors show that the epsilon proofreading subunit of DNA polymerase III contacts the beta processivity clamp, which increases the stability of the alpha polymerase/beta clamp complex. The authors postulate that the interaction between epsilon and the beta clamp could prevent PolIV from binding to the clamp during active processive replication. The crosslinking and MS results provide an interesting approach that complements structure determination of the individual subunits in the alpha/beta/epsilon complex, providing a novel view of this central subcomplex within the replicative DNA polymerase III holoenzyme. While the model is generally supported, several concerns listed below need to be addressed to provide the confidence needed to accept the attractive overall model.

Major points:

1. The authors neglect to show any sizing standards for the column chromatography experiments. In addition they only conduct the chromatography experiments on the protein complexes. Independent experiments with both protein sizing standards and individual proteins are necessary to complete these results.

These are now shown in Supplementary Figure S1A.

2. The authors posit a direct interaction between the beta clamp and epsilon but do not test this interaction directly with individual purified proteins in the absence of alpha. This would help to rule out an alternative explanation for the results in which epsilon indirectly stabilizes the alpha/beta complex. Such an indirect model persists since only a very small number of crosslinks (2) were found between epsilon and beta.

The direct interaction between the clamp and exonuclease as well as the lack of interaction between the clamp and a mutant of the exonuclease is now shown in Figure 3.

3. The experiment in Figure 3A shows a very minor difference in the migration of the clamp in the wild-type versus mutant Exo background. The authors should verify that this difference is real using their MS method.

We have replaced figure 3A with a figure showing the direct interaction between clamp and exonuclease. We believe that this provides the best proof of a direct interaction between the clamp and exonuclease.

4. The authors quantify the protein band intensity from sequential fractions from the column chromatography experiments yet they do not have error bars on their results. Were these experiments done multiple times? Error estimates need to be shown to allow the reader to determine the confidence in the results.

We have now included a measurement of relative amounts of bound and free clamp by fitting a 'sum of two Gaussians' (representing bound and free clamp) to the migration pattern of the clamp (shown in Supplementary Figure 1C). The advantage of this method is that it is insensitive to any difference in sample loading, gel staining, or gel imaging, and can therefore be used to compare different runs. For the estimation of the affinities we have used a total of nine different curves (three repeats of each of the three different concentrations: 1.5, 5.0, 10 μ M) for Pol III-clamp and nine curves for Pol III-clamp-exonuclease.

5. The authors do not address whether DNA affects the stability of the interactions. One would expect DNA to enhance the apparent affinity of the polymerase/clamp complex due to localization effects and one wonders whether this would alter the role of epsilon.

Unfortunately, we did not observe any effect of DNA in either the migration pattern on a gel filtration column, or the crosslinking results. This is not surprising as Pol III has an extremely low affinity for DNA that cannot be measured by conventional methods such as band shift or fluorescence anisotropy. A comparison of DNA binding by Pol III and Pol IV is shown in Supplementary Figure 7.

Minor points:

1. Figure S1: the legend states that stars indicate fractions used for mass spec analysis but I cannot see stars on the image.

We apologize for the error. This is now corrected.

2. Figure S5: It would seem that a replication fork structure would be more appropriate for the gel shift experiment rather than double stranded DNA.

We apologize for not being clear about the DNA substrate. We have used a primed DNA substrate (25 base pair double stranded DNA with a 5 nucleotide single stranded overhang, fluorescently (Cy5) labelled on the 5' of the complementary strand). This is now indicated in the figure legend.

A replication fork is not the natural substrate for Pol III as this structure is encountered by the DNA helicase (DnaB), not the polymerase. However, out of interest we have also tested a forked DNA substrate (35 double stranded DNA with two, non-complementary 10 nucleotide overhangs) for Pol III and Pol IV binding. As expected, this structure shows no appreciable binding to Pol III while Pol IV binds the forked DNA with comparable affinity as a primed DNA substrate. This is shown in the figure below. Left panel shows the fluorescently labelled DNA, right panel shows the same gel after Coomassie brilliant blue staining of the proteins. Lanes 1+11: Pol III alone, 2+12: Pol III with primed DNA, 3+13: Pol III with forked DNA, 4+14: Pol IV alone, 5+15: Pol IV with primed DNA, 6+16, Pol IV with forked DNA.

3. Were the 6xhis tags removed from the proteins prior to the experiments? The authors should make a statement regarding any effects that they might expect from the tags.

The tags were not removed as we did not see any differences between tagged and un-tagged proteins. A comparison of the his-tagged and wild-type proteins is shown in Supplementary Figure S4.

Referee #2

Summary: The manuscript by Rêgo et al. probes interactions of the E. coli sliding clamp protein with the alpha (catalytic) and epsilon (proofreading) subunits of the main replicase, DNA polymerase III (Pol III), and with the specialized translesion synthesis Pol IV. Based on results of gel filtration and crosslinking experiments, the authors conclude that epsilon contacts the cleft of the clamp to regulate access of Pol IV to the replication fork. Although the proposed model is intriguing, the results, as presented, are qualitative rather than quantitative, making it difficult to gauge the significance of the proposed interaction. Related to this, the study also lacks functional studies that could speak to the biological relevance of the epsilon-clamp interaction. In addition, certain of the findings presented are confirmatory of earlier published works, reducing the overall novelty and impact of the work. Finally, the authors fail to discuss their findings in the context of previously published models for Pol III-Pol IV switching.

Major comments:

1. The authors have chosen to use gel filtration chromatography to characterize interactions of epsilon with the alpha and beta subunits of Pol III. Although they present evidence that epsilon interacts with the Pol III alpha-clamp complex, and appears to stabilize it, the results are qualitative in nature. Quantitative measure of the effect that wild type and mutant epsilon proteins have on the interaction are possible, and would help to greatly strengthen the authors' argument that epsilon contacts the clamp directly to stabilize the complex rather than influencing the affinity of Pol III alpha for clamp through an alpha-epsilon interaction. In addition, possible impact of other Pol III subunits (i.e., theta, which interacts with epsilon) on this interaction were not examined. Finally, the functional significance of the epsilon-clamp interaction is not examined. As a result, the importance of this interaction to Pol III function is unclear.

We have addressed all the points raised by the referee. We have quantified the interaction between the clamp and the other proteins as shown in Supplementary Figure 1C and discussed in the text. Furthermore, we now show that there is a direct interaction between the exonuclease and clamp and that mutation of the canonical clamp binding motif in the exonuclease abolishes the interaction. We have also examined the effect of the θ protein on the interaction between clamp, polymerase and exonuclease but found no observable difference (see Supplementary Figure S1B). Finally, we have now included a real-time primer extension assay to probe the functional significance of the exonuclease-clamp interaction (see Figure 4). Here we find that the direct interaction between the exonuclease and the clamp enhances the effectiveness of the exonuclease. Mutation of the clamp binding motif in the exonuclease renders the exonuclease less effective, to levels similar to that of exonuclease activity in the absence of the clamp.

2. The authors characterize the contributions of the "internal" and "C-terminal" clamp binding domains of Pol III alpha, and conclude that only the internal motif is involved in contacting the clamp. As noted by the authors, Dohrmann and McHenry (JMB, 350:228-239) presented compelling evidence that the internal clamp-binding motif of Pol III alpha was required for interaction with the clamp, while the C-terminal motif played an important role in contacting the tau subunit of Pol III. Although relevant to the story, the results presented here in regards to the internal and C-terminal clamp binding motifs are largely confirmatory of previously published works. Furthermore, on page 11, the authors discuss contributions of clamp surfaces other than the cleft to the clamp-Pol III alpha interactions. They may wish to discuss results from Scouten Ponticelli et al. (NAR, 37: 2796-2809), which presented evidence for additional contacts using a mutant clamp lacking functional clefts.

We agree with the referee that Dohrmann and McHenry 2005 convincingly showed that the C-terminal clamp binding motif is not essential for Pol III-clamp binding. However, three years later a crystal structure of the C-terminal clamp binding motif of Pol III bound to the clamp was published (Georgescu 2008), suggesting that it is involved in clamp binding. We therefore felt it was important to re-evaluate the function of the C-terminal clamp binding motif. Moreover, while Dohrmann provided biochemical data, we here present structure and sequence alignment based arguments that

further strengthen the case. Yet, because our conclusions are largely the same as in Dohrmann 2005, we have now shortened the piece. Furthermore, we have now also included a reference to (Scouten-Ponticelli 2009) in context of the multiple interactions between Pol III and clamp.

3. On page 12, and in the Discussion, the authors discuss models for Pol III-Pol IV switching, and the impact that the Pol III-epsilon-clamp interactions may have on the switch. However, these discussions do not take into account published models for the switch, including Indiani et al. (Mol. Cell, 19: 805-815), Furukohri et al. (JBC, 283: 11260-11269) and Wagner et al. (Mol. Microbiol., 74: 1143-1151), which proposed classical "tool belt" type models, and Heltzel et al. (PNAS, 106:12664-12669), which proposed a variation of the tool belt model involving a single clamp cleft. Presentation of quantitative and functional analyses of Pol IV interactions with different clamp-Pol III alpha-epsilon complexes would contribute significantly to this discussion.

We have now extended the discussion to include the mentioned publications and discuss our findings in the light of these earlier publications. We would also like to emphasize that our model for polymerase switching is described in the Discussion, where a degree of speculation is allowed, but that it stems from a logical combination of observations from our own results as well as earlier published work. Furthermore, we are of the opinion that the best way to investigate the interaction of Pol III-clamp –exonuclease and Pol IV is through means of single molecule studies, which are beyond the scope of these studies. To our knowledge all assays that have been used to study the switching of Pol III and Pol IV have been ‘bulk’ experiments, and as such confound in our view the different events that take place at the primers terminus.

Minor comments:

1. On page 4, the authors report the interaction between Pol III alpha and the clamp to be on the order of 1-10 μM . However, there are several reports that also conclude the interaction is sub-micromolar, including Kim and McHenry, JBC, 271: 20699-20704; Ponticelli et al., NAR, 37: 2796-2809.

Our quantification of the Pol III-clamp interaction shows that the K_D is $\sim 1.2 \mu\text{M}$. This is similar to Kim and McHenry, JBC, 271: 20699 ($K_D = 0.2-1 \mu\text{M}$), Dohrmann and McHenry JMB 350, 228-239 ($K_D = 0.8 \mu\text{M}$) and Lamers et al., Cell, 126, 881-892 ($K_D = 1.5 \mu\text{M}$). Scouten-Ponticelli et al., NAR, 37: 2796-2809, reported a 10-fold higher affinity for Pol III-clamp ($K_D = 0.1 \mu\text{M}$), but this difference may be explained by the different conditions and techniques used. Indeed, in the same publications the authors also measure the interaction between the clamp and δ by two different techniques (SPR and ITC) and find a 10-fold difference in K_D .

2. On page 7, the authors conclude, based largely on results of cross-linking experiments, that epsilon interacts directly with the clamp. Since these cross-linkers have spacer arms of $\sim 7-12$ angstroms, this conclusion strikes this reviewer as an overstatement.

We have adjusted the statement in the text from:

“... and that it interacts directly with the clamp.”

to:

Furthermore, the cross-linking results suggest a potential direct interaction between the exonuclease and the clamp. The interactions between the exonuclease and polymerase and clamp are further analysed below.

Referee #3

Exonuclease ϵ stabilizes the Pol III - clamp complex and prevents translesion DNA Pol IV access to the clamp

Ana Toste Rêgo, Andrew N. Holding, Helen Kent, and Meindert H. Lamers

In this manuscript the authors describe the structural characterization of a trimeric E.coli Pol III complex composed of the catalytic α , the DNA sliding clamp β and the exonuclease ϵ subunit. The identification of chemically cross-linked lysine residues by mass spectrometry indicated protein domains that are in close vicinity and allowed to delineate a subunit topology. Based on this

topology and previous observations the authors generated mutant subunit proteins to narrow down the binding interfaces and tested the effect of these mutations on protein-protein binding by size exclusion chromatography and co-elution of the interacting proteins. Gel filtration of the wild-type proteins indicated that the exonuclease enhances the binding of the catalytic to the clamp subunit which is in agreement with the cross-link data that position the exonuclease between the catalytic and clamp subunits. Cross-links further suggested that the exonuclease binds the PHP domain of the catalytic

subunits by wrapping its C-terminal tail around. Mutations of the clamp binding motifs of the catalytic and exonuclease subunit revealed that these motifs are important for the interaction of the catalytic with the clamp subunit. Truncations of the C-terminal tail of the catalytic subunit revealed that several sites within the C-terminus contribute to clamp binding. In addition, the authors showed that the catalytic subunit of Pol III competes with Pol IV for binding to the clamp subunit and that this effect is enhanced by the presence of the exonuclease. The authors suggested that the various sites of the catalytic and exonuclease subunit contributing to clamp binding provide a mechanism for the fine-tuning of the replication machinery that allows the removal of bases by the exonuclease and the repair of DNA lesions by Pol IV.

In general, the conclusions are in agreement with the experimental findings. It has to be pointed out that cross-links are rather "flexible restraints" that indicate spatial proximity between protein domains and facilitate to draw a topological map at low resolution. Consequently, the findings that cross-links indicate that the exonuclease is sandwiched by the catalytic and clamp subunits or that the exonuclease tail may be wrapped around the catalytic subunit provide a novel structural model that clearly needs further structural analysis by EM or crystallography. The reviewer thus argues for a more careful interpretation of the data by taking into account that a topology based on a few clusters of cross-links may be associated with significant inaccuracies.

We have been careful in the analysis of the cross-links and have set a high cut-off for accepting a match between the observed fragmentation pattern and the theoretical crosslinks. As a result, for the known part of the polymerase we find an excellent correlation between the observed crosslinks and the crystal structure. Furthermore, we find crosslinks between the Pol III and clamp that are in agreement with previously published data (Dohrmann and McHenry JMB 350, 228-239). For the potential novel interactions identified by our crosslinking approach we have used additional biochemical experiments to verify the observed crosslinks. As such, we have verified the interaction between the tail of the exonuclease and the PHP domain of Pol III using different deletion constructs of the polymerase. We have furthermore analysed the potential direct interaction between the exonuclease and clamp by gel filtration, mutagenesis and a primer extension assay. Hence, we are confident that the combination of the crosslinking and additional experiments provides an accurate model of the Pol III-clamp-exonuclease complex.

Major comments

1. The cross-links identified by mass spectrometry are crucial for the design of the binding experiments and the interpretation of the results. Based on the information shown in the manuscript it is impossible to judge the quality of the cross-linking analysis. The reviewer thus requests detailed information on the cross-link identifications. As a minimum requirement the authors have to provide the sequences of the cross-linked peptides indicating the linked lysines and the precursor mass error. Furthermore, the authors have to state the minimum number of fragment ions per peptide that was considered to be sufficient for confident cross-link identifications. Likewise, the reviewer recommends to provide labeled fragment ion spectra of cross-links that are important for the conclusions of this study.

As requested we now provide a summary of the sequence and precursor mass error for each cross-linked position in Supplementary Figure 3B. For clarity, where multiple versions of the same cross-links were detected only one sequence has been provided. Additionally, we have provided the precursor mass:charge ratio, and charge. The identified peptides were scored probabilistically by matching fragmentation ions using an algorithm adapted from previously described linear peptide analysis scoring techniques (Cox et al, J. Proteome Res. 10: 1794-1805 and Holding et al. 2013, manuscript in preparation). A minimum score of 250 was used as a cut-off in identifying peptides. This figure was calculated as $-10 \times \text{Log}_e$ (the probability that the identified number of theoretical ions or more was matched to the fragmentation spectra by chance). This was found to be a more stringent method to counting the number of fragment ions per peptides alone. Using this method we find a minimum of 20 ions per peptide.

2. The authors state that the cross-links are in agreement or do not violate the structural model, however, they do not explain what measure they apply to judge violation of the cross-linking results. The authors should explain how they calculate this distance measure and whether there is also some flexibility associated with this value.

*We have now included an explanation of the distance measure in the manuscript (page 7):
“The average observed distance between the C α atom of two cross-linked lysines is ~22 Å, which is well within the predicted distance of 24 Å (2 x length of a lysine side chain (6.4Å) + length of the cross linker BS3 (11.4Å)). We also find a few longer cross-links, with some distances reaching 28Å. However, these distances are measured on a static crystal structure and are not taking into account any molecular motions of the protein, which are substantial in Pol III and DNA polymerases in general (see (Steitz & Yin, 2004; Wing et al, 2008; Evans et al, 2008)). Similarly, distances of up to 28Å were also observed in (Chen et al, 2010).”*

3. On page 5 it is indicated that the authors employed a cross-linking approach similar to Chen et al. In contrast to this study, Chen et al. did not use isotopically labeled cross-linkers. With the exception of the strong cation exchange chromatography for the enrichment of cross-linked peptides there are no similarities to the strategy described by Chen et al. In case there are no suitable references describing this approach, the authors have to explain their strategy and the search engine used to identify the cross-links in more detail.

The use of isotopic labelled cross-linking reagents was previously described by Leitner et al., Mol Cell Proteomics, 9: 1634-1649, and has now been referenced appropriately. Our search engine utilizes a previously described algorithm (Cox et al., J. Proteome Res., 10, 1794-1805) that has been adapted for cross-link peptide identification (Holding et al., manuscript in preparation 2013).

In its present form the manuscript is not recommended for publication in EMBO Journal.

Minor comments:

1. The authors used several times the term "affinity".

On page 4 "the affinity between Pol III and clamp is rather low, between 1-10 μ M": μ M rather refers to the KD.

This has been changed to:

“the affinity between Pol III and clamp is rather low, with reported values for K_D of ~1 μ M”

In several other instances the authors used the term "affinity", however, were not measuring affinity but assessed enhanced/decreased binding or interaction based on co-elution from the gelfiltration column.

Interaction is a more general term referring to how two substances may affect one another. Affinity is specific for the binding of two substances, or “the degree to which a substance tends to combine with another” (Oxford Dictionary) and is therefore appropriately used throughout the manuscript. In turn, K_D is a measure of affinity.

2. Page 5, line 23: "(Chen et al, 2009)" change to "(Chen et al, 2010)"

This has been corrected

3. Page 8, line 24: "extend" change to "extent"

This sentence has been replaced by:

“Following this, we first measured the direct interaction between the exonuclease and the clamp (Figure 3). This interaction is rather weak and even at 30 μ M we only observe a part of the exonuclease co-migrate with the clamp. In contrast, the mutant exonuclease^{185/187} has lost all affinity for the clamp and travels unaltered at all three concentrations, providing proof that the exonuclease directly binds to the clamp using a canonical binding motif.”

4. Page 11, line 11: The acronym "OB" has not been introduced.

The full name of the OB domain has now been added to the text:

“... the oligonucleotide/oligosaccharide binding (OB) domain ...”

5. Page 12, line 17: "competes of Pol IV" change to "competes off Pol IV"
This has been corrected in the text.

6. Page 16, line 15: "reverse-phase" change to "reversed-phase"
This has been corrected in the text.

At the author's/editor's discretion

It remains within the editor's/author's responsibility to decide whether cross-links identified by a software that has not yet been published are acceptable.

2nd Editorial Decision

26 February 2013

Thank you for submitting your revised manuscript for our consideration. It has now been re-reviewed by one of the original reviewers (see comments copied below), and I am happy to inform you that this referee considers the study significantly improved and now in principle suitable for publication, pending minor alterations of a few textual/discussion points. In addition, there are several editorial points that I would ask you to address before acceptance:

- please consider the suggestions of referee 2 and incorporate them into the manuscript where appropriate; also please briefly respond/comment on each of these points in a final response letter to the editor.

- please add a brief 'Author Contribution' description at the end of the manuscript text (next to the 'Acknowledgement' and 'Conflict of Interest' sections).

- please carefully check for complete formatting of all references - I noticed some incompletely converted Endnote references e.g. on the bottom of page 3.

- in several figures, resolution is too low and resulting in pixelated images for parts of them. This concerns the small gel insets in Figures 1/3/6/7, as well as the structural models in Figure 2. For production purposes, please upload revised figures with less pixelated appearance in these panels.

- finally, I wanted to discuss possible changes to the title. I appreciate that the current title puts strong emphasis the biological/functional insights derived from your analysis. However, this unfortunately does not allow to mention the important aspect of structural/architectural analysis through the crosslinking/mass-spec approach, which could lead to the paper being 'missed' by part of its target readership. Obviously the title shouldn't sound too 'technical', and we also want to avoid it turning into a 3-line monster, but maybe you could imagine some way of combining the two key aspects of this study into one concise sentence? Something along the lines of 'Architecture of the Pol III-clamp-exonuclease complex reveals...'? Please note that we have some flexibility to go over the 100-character limit stated in our guidelines in such cases.

I am thus returning the manuscript to you for a final round of minor revision, in order to allow you to make these last modifications. I would appreciate if you could send in your re-revised study as soon as possible, ideally within one week, and already include the signed relevant publication license forms (see below) at this stage. Please do not hesitate to get back to me should you have any questions in the meantime, or if you would like to discuss possible alternative titles.

I look forward to receiving your final version.

REFEREE REPORTS

Referee #2

The authors have done a solid job of addressing the prior round of comments raised by reviewers. This revised work is significantly stronger, and presents a compelling case for epsilon-clamp interactions. I have only a few minor comments that the authors should consider. These comments

are intended to clarify what is already a very strong presentation.

- 1) On pg. 8, line 3, the authors state "...indicating that the tail fits..." The authors may want to relax this statement, since cross-linking can trap complexes that are transient as well as stable. Perhaps "suggests"?
- 2) On pg. 11, line 8, the authors state "...but also makes the exonuclease more efficient." Do the authors mean "positions" instead of "makes"? Makes implies an effect on catalysis, while their argument seems to be that of a positional effect.
- 3) On page 13, line 8, the authors refer to the 917 deletion mutant as "...having lost all clamp binding regions." How can this be known if the interactions have yet to be fully mapped?
- 4) On pg. 13, line 21, the authors refer to interactions of Pol IV with stalled replication machinery. I believe these interacts involved just clamp, pol III alpha, and Pol IV in solution, in the absence of DNA, nucleotides, etc.
- 5) The authors might want to consider a more extensive discussion of possible effects of DNA on the interactions they have studied. They indicated they did not notice an effect of DNA on the clamp-epsilon interaction, but it nevertheless seems likely that DNA is impacting the coordination of Pol III alpha, epsilon, clamp and Pol IV.
- 6) On page 15, line 11, the authors note that cited studies observed displacement of Pol III from the primer-template junction by >50-fold levels of Pol IV. I believe that both the Indiani and Heltzel references saw Pol IV takeover at significantly lower ratios (as low as 1- to 2-fold).

2nd Revision - authors' response

27 February 2013

Editorial points:

-Please consider the suggestions of referee 2 and incorporate them into the manuscript where appropriate; also please briefly respond/comment on each of these points in a final response letter to the editor.

Our responses to the points raised by referee 2 are detailed below the editorial points.

- Please add a brief 'Author Contribution' description at the end of the manuscript text (next to the 'Acknowledgement' and 'Conflict of Interest' sections).

This has now been included.

- Please carefully check for complete formatting of all references - I noticed some incompletely converted Endnote references e.g. on the bottom of page 3.

This has been corrected. No other unformatted references were found in the text.

- In several figures, resolution is too low and resulting in pixelated images for parts of them. This concerns the small gel insets in Figures 1/3/6/7, as well as the structural models in Figure 2. For production purposes, please upload revised figures with less pixelated appearance in these panels.

The low resolution of the figures was due to the conversion to .pdf format to limit the file size. The original Adobe Illustrator files contain high resolution figures and are submitted with the revised manuscript.

- Finally, I wanted to discuss possible changes to the title. I appreciate that the current title puts strong emphasis the biological/functional insights derived from your analysis. However, this unfortunately does not allow to mention the important aspect of structural/architectural analysis through the crosslinking/mass-spec approach, which could lead to the paper being 'missed' by part of its target readership. Obviously the title shouldn't sound too 'technical', and we also want to avoid it

turning into a 3-line monster, but maybe you could imagine some way of combining the two key aspects of this study into one concise sentence? Something along the lines of 'Architecture of the Pol III-clamp-exonuclease complex reveals...'? Please note that we have some flexibility to go over the 100-character limit stated in our guidelines in such cases.

As we discussed by email we have now changed the title to:

“Architecture of the Pol III-clamp-exonuclease complex reveals key roles of the exonuclease subunit in processive DNA synthesis and repair”

Referee #2

The authors have done a solid job of addressing the prior round of comments raised by reviewers. This revised work is significantly stronger, and presents a compelling case for epsilon-clamp interactions. I have only a few minor comments that the authors should consider. These comments are intended to clarify what is already a very strong presentation.

1) On pg. 8, line 3, the authors state "...indicating that the tail fits..." The authors may want to relax this statement, since cross-linking can trap complexes that are transient as well as stable. Perhaps "suggests"?

The text has been changed from “indicating that the tail of Pol III” to: “suggesting that the tail of Pol III”

2) On pg. 11, line 8, the authors state "...but also makes the exonuclease more efficient." Do the authors mean "positions" instead of "makes"? Makes implies an effect on catalysis, while their argument seems to be that of a positional effect.

The text has been changed from “but also makes the exonuclease more efficient” to: “but also positions the exonuclease in a conformation that is more favorable for DNA access”

3) On page 13, line 8, the authors refer to the 917 deletion mutant as "...having lost all clamp binding regions." How can this be known if the interactions have yet to be fully mapped?

Although all the details of the contacts between Pol III and clamp are still not known, it is clear that all clamp interaction regions reside in the last ~240 residues of the polymerase. Removal of the last ~240 residues abolishes all interactions between Pol III and clamp (Figure 6E). In addition, in Lamers et al (Cell 126, 881-892) we report the same result. This reference has now been added to the text.

Nonetheless we have changed the text somewhat from: “as this construct has lost all clamp binding regions” to: “suggesting that this construct has lost all clamp binding regions”.

4) On pg. 13, line 21, the authors refer to interactions of Pol IV with stalled replication machinery. I believe these interactions involved just clamp, pol III alpha, and Pol IV in solution, in the absence of DNA, nucleotides, etc.

The text has been changed to:

“Yet it has been reported that Pol III and the translesion (TLS) DNA polymerase Pol IV can bind to the clamp simultaneously (Indiani et al, 2005) and that the two polymerases switch positions on the DNA during stalling of the replication machinery, but not during active replication...”

5) The authors might want to consider a more extensive discussion of possible effects of DNA on the interactions they have studied. They indicated they did not notice an effect of DNA on the clamp-epsilon interaction, but it nevertheless seems likely that DNA is impacting the coordination of Pol III alpha, epsilon, clamp and Pol IV.

A more ‘extensive discussion of possible effects of the DNA’ is complicated by the fact that little is known about the interaction between Pol III and DNA or what happens when Pol III switches from rapid DNA synthesis to stalling. Therefore a discussion on this subject will be rather speculative. However, we have added a final sentence to our discussion that places more emphasis on the role of

the DNA: "Thus our model predicts that it is the structure of the DNA template rather than the protein itself that dictates which activity is selected: rapid DNA synthesis by Pol III, nucleotide removal by the exonuclease, or lesion bypass by Pol IV."

6) On page 15, line 11, the authors note that cited studies observed displacement of Pol III from the primer-template junction by >50-fold levels of Pol IV. I believe that both the Indiani and Heltzel references saw Pol IV takeover at significantly lower ratios (as low as 1- to 2-fold).

While different degrees of inhibition of Pol III DNA synthesis are observable at lower concentrations of Pol IV, complete inhibition of Pol III holoenzyme activity requires significantly higher concentrations of Pol IV. Unfortunately, different reports show varying degrees of molar excess of Pol IV required to reach 100% inhibition: 25-fold excess (Indiani et al. (Mol. Cell, 19: 805-815), Heltzel et al. (PNAS, 106:12664-12669)) 80-fold excess (Wagner et al. (Mol. Microbiol., 74: 1143-1151)) and >100-fold excess (Furukhori et al. (JBC, 283: 11260-11269)). Our initial 50-fold excess was used as an indicative average number.

We have adjusted the text to make this clearer: "At higher concentrations, Pol IV is capable of displacing a stalled Pol III holoenzyme from the primer junction in vitro, ultimately resulting in a complete inhibition of replicative DNA synthesis (requiring a 25-100 fold molar excess of Pol IV)..."